

# Interactions between channels and tributary alluvial fans: channel adjustments and sediment-signal propagation

Sara Savi[1], Stefanie Tofelde[2,3], Andrew D. Wickert[4], Aaron Bufe[3], Taylor F. Schildgen[1,3], and Manfred R. Strecker[1]

[1]Institut für Geowissenschaften, Universität Potsdam, 14476 Potsdam, Germany
[2]Institut für Umweltwissenschaften und Geographie, Universität Potsdam, 14476 Potsdam, Germany
[3]Helmholtz Zentrum Potsdam, GeoForschungsZentrum (GFZ) Potsdam, 14473 Potsdam, Germany
[4]Department of Earth Sciences and Saint Anthony Falls Laboratory, University of Minnesota, Minneapolis, MN 55455, USA

Corresponding Author: Sara Savi (savi@geo.uni-potsdam.de)

## Abstract

Climate and tectonics impact water and sediment fluxes to fluvial systems. These boundary conditions set river form and can be recorded by fluvial deposits. Reconstructions of boundary conditions from these deposits, however, is complicated by complex channel-network interactions and associated sediment storage and release through the fluvial system. To address this challenge, we used a physical experiment to study the interplay between a main channel and a tributary under different forcing conditions. In particular, we investigated the impact of a single tributary junction, where sediment supply from the tributary can produce an alluvial fan, on channel geometries and associated sediment-transfer dynamics. We found that the presence of an alluvial fan may promote or prevent sediment to be moved within the fluvial system, creating different coupling conditions. A prograding alluvial fan, for example, has the potential to disrupt

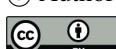



the sedimentary signal propagating downstream through the confluence zone. By analyzing
different environmental scenarios, our results indicate the contribution of the two sub-systems to
fluvial deposits, both upstream and downstream of the tributary junction, which may be
diagnostic of a perturbation affecting the tributary or the main channel only. We summarize all
findings in a new conceptual framework that illustrates the possible interactions between
tributary alluvial fans and a main channel under different environmental conditions. This
framework provides a better understanding of the composition and architecture of fluvial
sedimentary deposits found at confluence zones, which is essential for a correct reconstruction of
the climatic or tectonic history of a basin.

## 1. Introduction

The geometry of channels and the downstream transport of sediment and water in rivers are

determined by climatic and tectonic boundary conditions (Allen, 2008, and references therein).
Fluvial deposits and landforms such as conglomeratic fill terraces or alluvial fans may record
phases of aggradation and erosion that are linked to changes in sediment or water discharge, and
thus provide important archives of past environmental conditions (Armitage et al., 2011;
Castelltort and Van Den Driessche, 2003; Densmore et al., 2007; Mather et al., 2017; Rohais et
al., 2012; Tofelde et al., 2017). Tributaries are an important component of fluvial networks, but
their contribution to the sediment supply of a river channel can vary substantially (Bull, 1964;
Hooke, 1967; Lane 1955; Leopold and Maddock, 1953; Mackin, 1948; Miller, 1958). Their
impact on the receiving river (referred to as *main channel* hereafter) may not be captured by
numerical models of alluvial channels, as most models either parameterize the impacts of
tributaries into simple relationships between drainage-basin area and river discharge (Whipple
and Tucker, 2002; Wickert and Schildgen, 2019), or treat the main channel as a single channel
with no lateral input (e.g., Simpson and Castelltort, 2012). Extensive studies on river confluences
(e.g., Rice et al., 2008 and references therein) mainly focus on (1) hydraulic parameters of the
water flow dynamics at the junction (Best 1986, 1988), which are relevant for management of
infrastructure (e.g., bridges), and (2) morphological changes of the main channel bed, which are
relevant for sedimentological studies and riverine habitats (Benda et al., 2004a; Best 1986; Best





and Rhoads, 2008). Geomorphological changes (i.e., channel slope, width, or grain-size
distribution) have been studied in steady-state conditions only (Ferguson et al., 2006; Ferguson
and Hoey, 2008), and with no focus on fluvial deposits related to the interactions between
tributaries and the main channel. In source-to-sink studies an understanding of these processes,
however, is relevant for the reconstruction of the climatic or tectonic history of a certain basin.
By modulating the sediment supplied to the main channel, tributaries may influence the
distribution of sediment within the fluvial system and the origin and amount of sediment stored
within fluvial deposits at confluence zones. Additionally, complex feedbacks between tributaries
and main channels (e.g., Schumm, 1973; Schumm and Parker, 1973) may enhance or reduce the
effects of external forcing on the fluvial system, thus complicating attempts to reconstruct past
environmental changes from these sedimentary deposits.
The dynamics of alluvial fans can introduce an additional level of complication to the
relationship between tributaries and main channels. Fans retain sediment from the tributary and
influence the response of the connected fluvial system to environmental perturbations (Ferguson
and Hoey, 2008; Mather et al., 2017). Despite the widespread use of alluvial fans to decipher
past environmental conditions (Bull, 1964; Colombo et al., 2000; D'Arcy et al., 2017; Densmore
et al., 2007; Gao et al., 2018; Harvey, 1996; Savi et al., 2014; Schildgen et al., 2016), we still
lack a clear understanding of the interactions between alluvial fans and main channels under the
influence of different environmental forcing mechanisms. The lack of a systematic analysis of
these interactions represents a major gap in knowledge that hinders our understanding of (1) how
channels respond to changes in water and sediment supply at confluence zones, and (2) how
sediment moves within fluvial systems (Mather et al., 2017), with potential consequences for
sediment-transport dynamics as well as the composition and architecture of fluvial sedimentary
deposits.
In this study, we analyze the interplay between a main channel and a tributary under different
environmental forcing conditions in an experimental setting, with particular attention to
tributaries that generate an alluvial fan. Physical experiments have the advantage of providing a
simplified setting with controlled boundary conditions and that may include water and sediment
discharge, and uplift rate or base-level changes. These models may thus capture many



components of complex natural behaviors (Hooke, 1967; Paola et al., 2009; Schumm and Parker,
1973), and they provide an opportunity to analyze processes at higher spatial and temporal
resolution than is generally possible in nature (e.g., De Haas et al., 2016; Parker, 2010; Reitz et
al., 2010). These characteristics allow us to directly observe connections between external
perturbations (e.g., tectonic or climatic variations) and surface processes impacting landscapes.

We present results from two groups of experiments in which we separately imposed a

perturbation either in the tributary only (Group 1, Fig. 1a, b) or solely in the main channel
(Group 2, Fig. 1c). Group 1 can be further subdivided into cases in which the tributary has: (a) an
aggrading alluvial fan (Fig. 1a); or (b) an incising alluvial fan (Fig. 1b), whereas Group 2
represents a case of a sudden increase in water discharge in the main channel (Fig. 1c). These
three cases represent what may occur in many natural environments (e.g., Hamilton et al., 2013;
Leeder and Mack, 2001; Mather et al., 2017; Schumm 1973; Van Djik et al., 2009).

By analyzing how a tributary may affect the main channel under these different forcing

conditions, we aim to build a conceptual framework that lends insight into the interplay between
alluvial fans and main channels. Toward this goal, we provide a schematic representation of how
the downstream delivery of sediment changes under different environmental conditions. Through
this representation, we hope to contribute to a better understanding and interpretation of fluvial
morphologies and sedimentary records, which may hold important information about regional
climatic and tectonic history (Allen, 2008; Armitage et al., 2011; Castelltort and Van Den
Driessche, 2003; Densmore et al., 2007; Mather et al., 2017; Rohais et al., 2012).

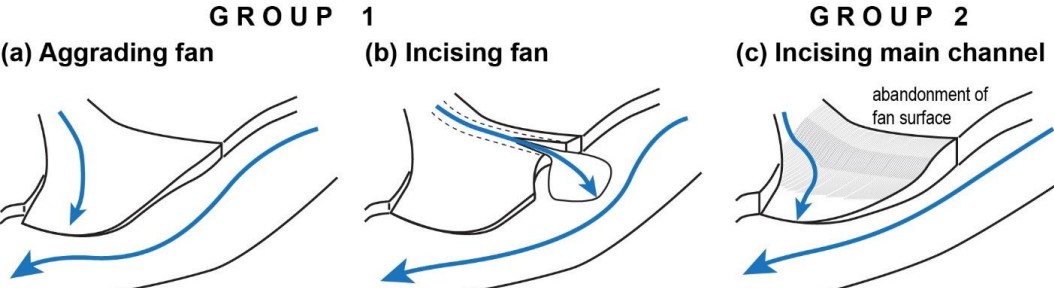






Figure 1. Schematic representation of the three scenarios analyzed in this study.

## 2. Background and Motivation

### 2.1. Geometry and sediment transfer dynamics in a single-channel system

#### 2.1.1. General concepts

An alluvial river is considered to be in steady state (*equilibrium regime*) when its water discharge provides sufficient power, or sediment-transport capacity, to transport the sediment load supplied from the upstream contributing area at a given channel slope (Bull, 1979; Gilbert, 1877; Lane, 1955; Mackin, 1948). When that power is insufficient, sediment is deposited within the channel (*aggradation*), whereas when the sediment-transport capacity exceeds the sediment supply, the river erodes the channel banks and bed (*incision*) (Lane, 1955). Any change in sediment or water supply modifies the sediment-to-water ratio, such that the river must transiently adjust one or more of its geometric features (e.g., slope, width, depth, or grain-size distribution) to re-establish equilibrium (Mackin 1948; Meyer-Peter and Müller, 1948).

When a perturbation occurs in the system, slope adjustments are not uniform along the channel. If the perturbation occurs upstream (e.g., in water or sediment supply), channel slope changes first at the channel head through incision or aggradation (e.g., Simpson and Castelltort 2012; Tofelde et al., 2019; Van den Berg Van Saparoea and Potsma, 2008; Wickert and Schildgen, 2019). With time, slope adjustments proceed downstream until the entire channel slope has adjusted to the new condition. Conversely, when perturbations occur downstream (e.g., a change in base level), the slope initially changes at the channel mouth, and the slope adjustment propagates upstream until the entire channel is adjusted to the new base level (Parker et al., 1998; Tofelde et al., 2019; Van den Berg Van Saparoea and Potsma, 2008; Whipple et al., 1998).

At the scale of a drainage network, these geometric adjustments may alter the mechanisms and rates at which sediment is moved across landscapes. In general, under both steady and transient conditions, sediment moves from zones of erosion to areas of deposition passing





through a *transfer zone* (Castelltort and Van Den Driessche, 2003). The capacity of the transfer
zone to temporarily store or release sediment can influence the amount and the provenance of
sediment reaching the depositional zone, buffering the sedimentary signal carried through the
system (Tofelde et al., 2019). This buffering may be particularly important for the outcome of
analyses that use the geochemical composition of sediment (e.g., cosmogenic nuclide
concentrations) to date fluvial deposits or infer changes in erosion rate (Biermann and Steig,
1996; Granger et al., 1996, Lupker et al., 2012; Wittmann and von Blanckenburg, 2009;
Wittmann et al., 2011).
Although our understanding of buffering within the sediment-transfer zone helps to explain
how landscape perturbations are recorded in river morphology and downstream sedimentary
records, to date neither physical (Tofelde et al., 2019), theoretical (Castelltort and Van Den
Driessche, 2003; Paola et al., 1992; Wickert and Schildgen, 2019), nor numerical (Simpson and
Castelltort, 2012; Wickert and Schildgen, 2019) models take into account how the dynamics of
tributary junctions affect the geometry or sediment transport of the main channel. Tributary sub-
systems exist across spatial scales from small headwater catchments to continental-scale rivers
(i.e., short to large transfer zones). They may alter the amount of sediment entering the transfer
zone, modifying the sediment-input signals that can be recorded by fluvial terraces and
sedimentary basins. Understanding how tributaries and their fans interact with the main channel
is critical to correctly reconstruct external forcing conditions from the sediments of alluvial fans,
fluvial terraces, and depositional sinks.
*2.1.2.   Alluvial fans*
Alluvial fans typically form at points of rapid decrease of channel slope and/or increases in
valley width (Benda, 2008; Bull, 1964). Their depositional processes are characterized by a
combination of sheet flows and channelized flows that are interrupted by large reorganizations of
the channel system through avulsions (Bryant et al., 1995; De Haas et al., 2016; Hooke and
Rohrer, 1979; Reitz et al., 2010; Reitz and Jerolmack, 2012). Variations in these processes can
be related to the internal, i.e. autogenic, dynamics of the system (Hamilton et al., 2013; Kim and
Jerolmack, 2008; Van Djik et al., 2009, 2012) or to external forcings (Armitage et al., 2011;
Rohais et al., 2012). In general, sheet flows deposit sediment uniformly over the entire fan

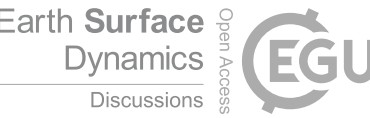

surface. Conversely, channelization on fans is generally associated with localized erosion.
Avulsions are sudden reorganizations of the channel system that are integral to the cyclic
construction of a fan (Straub et al., 2009). They occur when channels aggrade above the fan
surface and suddenly change position to start deposition on a new location of the fan surface
(Hamilton et al., 2013; Van Djik et al., 2009).
In our experiments, we distinguish between two modes of fan construction: *fan aggradation*,
i.e., deposition of material on the fan surface, which leads to an increase in the fan surface
elevation, and *fan progradation*, i.e., deposition that occurs at the downstream margin of the fan,
which leads to fan lengthening. Progradation may occur during both aggradation and incision
phases (Fig. 1).
## 2.2. Geometry and sediment-transfer dynamics in a multi-channel system
### 2.2.1. Tributary influence on main channel
At confluence zones, the main channel is expected to adapt its width, slope, transport rate,
and sediment-size distribution according to the combined water and sediment supply from the
main channel and the tributary (Benda et al., 2004b; Best, 1986; Ferguson et al., 2006; Lane
1955; Miller, 1958; Rice et al., 2008). Consequently, a perturbation occurring in the tributary
will also affect the main channel. For example, a sudden increase in sediment input from a
tributary (e.g., from a landslide or debris flow) can overwhelm the transport capacity of the main
channel, thereby inducing sediment deposition at the confluence (Fig. 1a). As a result, the main
channel upstream of the tributary experiences a rise in its local base level, which causes
additional local deposition and a transient reduction in the main-channel slope upstream of the
tributary (Ferguson et al., 2006; Benda, 2008; Benda et al., 2004b). This sediment deposition
upstream from the tributary increases the slope of the main channel downstream of the tributary,
until the main channel is adjusted to transporting the higher sediment load (Benda et al., 2003;
Ferguson et al., 2006; Ferguson and Hoey, 2008; Mackin, 1948; Rice and Church, 2001). It
follows that the main channel both upstream and downstream from the tributary should undergo
an aggradation phase, the former due to an increase in its local base level at the junction, the
latter because of an increase in sediment supply from the tributary (Ferguson and Hoey, 2008;
Mackin, 1948; Rice and Church, 2001). In their numerical model, Ferguson et al. (2006) found





that when tributaries cause aggradation at the junction with the main channel, the main channel
slope adjustments extend approximately twice as far upstream as they do downstream. They
additionally found that variations in grain-size input from a tributary influence the grain-size
distribution in the main channel, both upstream and downstream of the tributary junction.
Considering that in our experiments we used a homogeneous grain size, the work of Ferguson et
al. (2006) complements our analyses.

Whether the tributary is aggrading, incising, or in equilibrium may also have important

consequences for *how* and *where* local fluvial deposits (i.e., alluvial-fan deposits or fluvial
terraces) reflect environmental signals. For example, when sediment is trapped within a
tributary's alluvial fan, the fan acts as *buffer* for the main channel, and environmental signals do
not propagate from the tributary into the fluvial deposits of the main channel (Ferguson and
Hoey, 2008; Mather et al., 2017). In contrast, where the tributary and main channel are fully
*coupled* (i.e. all sediment mobilized in the tributary reaches the main channel), the signal
transmitted from the tributary can be recorded in the stratigraphy of the main river (Mather et al.,
2017). Hence, to correctly interpret fluvial deposits and to reduce ambiguity, an understanding of
the aggradational/incisional state of the tributary and how this state influences the main channel
is important. In this study, we aim to provide this information for different tributary states.

*2.2.2.  Main channel influence on tributary*

The main channel influences a tributary primarily by setting its local base level. Therefore, a

change in the main-channel bed elevation through aggradation or incision represents a
downstream perturbation for the tributary, and tributary-channel adjustments will follow a
*bottom-up* propagation direction (Mather et al., 2017; Schumm and Parker, 1973). Typically, a
lowering of the main channel produces an initial phase of tributary-channel incision (Cohen and
Brierly, 2000; Fulkner et al., 2016; Germanoski and Ritter, 1988; Heine and Lant, 2009; Ritter et
al., 1995; Simon and Rinaldi, 2000), followed by channel widening (Cohen and Brierly, 2000;
Germanoski and Ritter, 1988), which occurs mainly through bank erosion and mass-wasting
processes (Simon and Rinaldi, 2000). As base-level lowering continues, the fan may become
entrenched, with the consequent abandonment of the fan surface and renewed deposition at a
lower elevation (Clark et al., 2010; Mather et al., 2017; Mouchené et al., 2017; Nicholas et al.,



2009) (Fig. 1c). In contrast, aggradation of the main channel may lead to tributary-channel
backfilling and avulsion (Bryant et al., 1995; De Haas et al., 2016; Hamilton et al. 2013; Kim
and Jerolmack, 2008; Van Djik et al., 2009, 2012).
When a non-incising main channel (*non-incising main axial river* of Leeder and Mack, 2001)
is characterized by efficient lateral erosion, it can efficiently erode the fan downstream margin,
thereby "cutting" its toe (Larson et al., 2015) (*fan-toe cutting* hereafter) (Fig. 1b). This toe-
cutting shortens the fan and increases the tributary channel slope. As a consequence, the increase
in transport capacity in the tributary triggers an upstream-migrating wave of incision. Fan-toe
cutting may thus cause fan incision and a consequent increase in sediment supply from the
tributary to the main channel (*healing wedge* hereafter; Leeder and Mack, 2001), in a process
similar to that caused by an incising main channel (*incising main axial river of* Leeder and Mack,

2001).

### 2.2.3.   Main channel and tributary interactions

Changes that occur in the tributary as a consequence of incision of the main channel may
alter the sediment supplied to the main river and create a series of autogenic feedback processes
that are generally referred to as a *complex response* (Schumm, 1973; Schumm and Parker, 1973).
These processes may form landforms such as cut-and-fill terraces that are not directly linked to
the original perturbation (Schumm, 1973), thereby complicating the reconstruction of past
environmental changes from such landforms. In our experiments, we analyze the changes
occurring in a tributary during a phase of main-channel incision to evaluate these potential
feedbacks.

## 3. Methods

### 3.1. Experimental setup

We conducted physical experiments at the Saint Anthony Falls Laboratory (Minneapolis,
USA). The experimental setup consisted of a wooden box with dimensions of 4 m x 2.5 m x 0.4
m, which was filled with quartz sand with a mean grain size of 144 μm (standard deviation of 40
μm). Two separate water and sediment input zones were used to form a main channel (MC) and


Earth **Surface**
**Dynamics**
Discussions
EGU

a tributary channel (T) (Fig. 2a). The main channel's input zone was located along the short side
of the box, whereas the tributary's input zone was located along the long side at a distance of 1.7
m downstream of the main-channel inlet (Fig. 2a). For each of the two input zones, the water
supply ($Q_w$) and sediment supply ($Q_{s\_in}$) could be regulated separately, and sand and water were
mixed before entering the box by feeding them through cylindrical wire-mesh diffusers filled
with gravel. Before entering the mesh, water was dyed blue to be visible on photos. At the
downstream end, sand ($Q_{s\_out}$) and water exited the basin through a 20 cm-wide gap that opened
onto the floor below. This downstream sink was required to avoid deltaic sediment deposition
that would, if allowed to grow, eventually raise the base level of the fluvial system. At the
beginning of each experiment, an initial channel was shaped by hand to allow the water to flow
towards the outlet of the box.

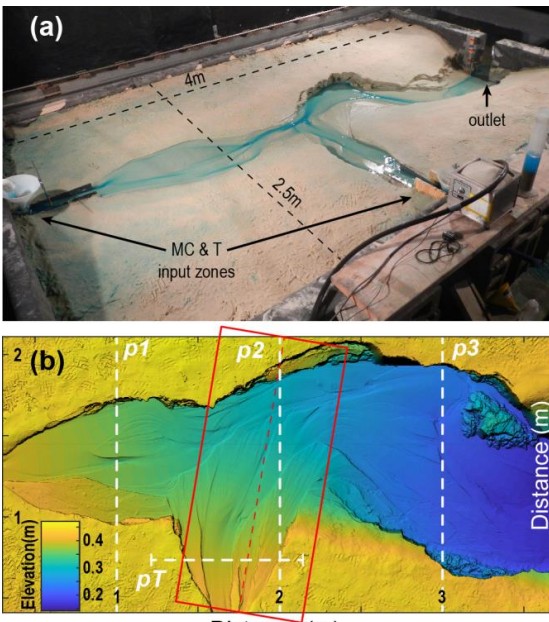


Figure 2. Experimental set-up. (a) Wooden box for the experiments showing the two zones of
sediment and water input, and the outlet of the basin. (b) Digital elevation model constructed
from laser scans (1 mm horizontal resolution). Red box shows the area of the swath grid used for
the calculation of the tributary long profile (Fig. 4) and slope values. Dashed white lines
represent the location of the cross sections shown in Figs. 5 and 6.




### 3.2. Boundary conditions

We performed six experiments with different settings and boundary conditions to simulate different tributary–main-channel interactions. As a reference, we included one experiment without a tributary and with a constant $Q_{s\_in}$ and $Q_w$ (MC_NC, where MC stands for *Main Channel only* and the suffix NC stands for *No Change* in boundary conditions; reported in Tofelde et al., 2019 as the Ctrl_2 experiment). The other five experiments all have a tributary and are divided into two groups: In Group 1, $Q_w$ and $Q_{s\_in}$ on the main channel were held constant, whereas we varied these inputs to the tributary. In Group 2, $Q_w$ and $Q_{s\_in}$ on the tributary were held constant, whereas we increased $Q_w$ in the main channel. In natural systems, changes in water and sediment supply may affect the main channel and tributary simultaneously, but to isolate the effects of the main channel and the tributary on each other, we studied perturbations that only affect one of them at a time. Our results can be combined to predict the response to a system-wide change in boundary conditions.

Each group includes one experiment with no change (NC) in $Q_{s\_in}$ and $Q_w$ (T_NC1 and T_NC2, where T stands for *run with Tributary* and the numbers at the end correspond to the group number). Group 1 includes one experiment with an increase followed by a decrease in $Q_{s\_in}$ in the tributary (T_ISDS, where ISDS stands for *Increasing Sediment Decreasing Sediment*) and one experiment with a decrease followed by an increase in $Q_w$ in the tributary (T_DWIW, where DWIW stands for *Decreasing Water Increasing Water*). Changes were first made in the direction that favored sediment deposition and the construction of an alluvial fan. Group 2 includes one experiment with no change (T_NC2) and one with an increase in $Q_w$ in the main channel (T_IWMC, where IWMC stands for *Increasing Water in Main Channel*). Importantly, the initial settings of the two groups of experiments are different (Table 1). In particular, initial $Q_w$ and $Q_{s\_in}$ of Group 2 guarantee a higher $Q_s/Q_w$ ratio, so that we could evaluate the effects of a change in the main-channel regime (from a *non-incising main river* to an *incising main river*) on the tributary and on sediment-signal propagation. In the context of this coupled tributary–main-channel system, we explore: 1) the geometric variations that occur in the main channel and in the



tributary (e.g., channel slope and valley geometry); and 2) the downstream delivery of sediment
and sedimentary signals.
**Table 1.** Overview of input parameters.

| EXP NAME | Initial conditions | | | | 1st change | | | 2nd change | |
|---|---|---|---|---|---|---|---|---|---|
| | **MC** | | **T** | | **MC** | **T** | | **T** | |
| | Qw | Qs_in | Qw | Qs_in | Qw | Qw | Qs_in | Qw | Qs_in |
| | mL/s | mL/s | mL/s | mL/s | mL/s | mL/s | mL/s | mL/s | mL/s |
| MC_NC** | 95 | 1.3 | | | | | | | |
| *Non-incising mean axial rivers – Group1* | | | | | *(at 300 min)* | | | *(at 375\* or 480 min)* | |
| T_NC1 | 95 | 1.3 | 63 | 2.2 | | | | | |
| T_ISDS | 95 | 1.3 | 63 | 2.2 | | | 4.5 | | 2.2 |
| T_DWIW* | 95 | 1.3 | 63 | 2.2 | | 31.5 | | 63 | |
| *Incising mean axial rivers - Group2* | | | | | *(at 180 min)* | | | | |
| T_NC2 | 63 | 1.3 | 41.5 | 2.2 | | | | | |
| T_IWMC | 63 | 1.3 | 41.5 | 2.2 | 126 | | | | |

* In the T_DWIW run the boundary condition change occurred at 375 min rather than 480 min
as in the T_ISDS experiment because fast aggradation that occurred at the tributary input zone
risked to overtop the wooden box margins.
**, Experiment published by Tofelde et al. (2019).

### 3.3. Measured and calculated parameters

*3.2.1.   Long profiles, valley cross-sections, and slope values*
Every 30 min we stopped the experiments to perform a scan with a laser scanner mounted
on the railing of the basin that surrounded the wooden box. Digital elevation models (DEMs)
created from the scans have a resolution of 1 mm (Fig. 2b). We extracted long profiles and valley
cross sections from these DEMs (i.e., elevation profiles perpendicular to the main flow direction)
for the main channel and the tributary. Long profiles for the main channel were calculated by
extracting the lowest elevation point along each cross section along the flow direction. Long
profiles for the tributary were calculated with a similar procedure using outputs from





Topotoolbox's SWATH profile algorithm (Schwanghart and Scherler, 2014) at 1 mm spatial
resolution along the line of the average flow direction (Fig. 2b). By plotting elevation against
down-valley or down-fan distance, rather than along the evolving path of the channels, the
resulting slopes are slightly overestimated due to the low sinuosity of the channels. Cross
sections were extracted at fixed positions, perpendicular to the main flow direction, for both the
main channel and the tributary (Fig. 2b).

For the main channel, spatially-averaged slopes were additionally calculated by manually

measuring the bed elevation at the inlet and at the outlet of the wooden box at 10-minute
intervals during the experiments. This procedure yielded real-time estimates of channel slope.
For comparison, spatially-averaged slopes where subsequently calculated also for the tributary
channel using the maximum and minimum elevation of the tributary long profile calculated
within the SWATH grid. Slope data are reported in the supplementary material.

*3.2.2.  Active valley-floor width and symmetry*

We defined the width of the active valley floor as the area along the main channel that was

occupied at least once by flowing water. It was measured along the main channel both upstream
and downstream of the tributary junction (Fig. 3a, upper panel). The active valley floor was
isolated by extracting all DEM values with an elevation of <0.42 m (where 0.42 m is the
elevation of the sand surface outside the manually-shaped channel) and with a slope of <15
degrees (a value visually selected from the DEMs as the best cut-off value for distinguishing the
valley floor from the banks). The average valley-floor width values were then calculated as the
average sum of pixels in each of the 700 cross sections within the selected zones (i.e., upstream
or downstream of the tributary junction; Fig. 3a, upper panel). The same method was used to
monitor valley symmetry. In this case, the averaged width was limited to the sum of pixels to the
left and to the right of an imaginary central line crossing the basin from the inlet to the outlet
(Fig. 3a). Small differences between left and right sums indicate high symmetry.





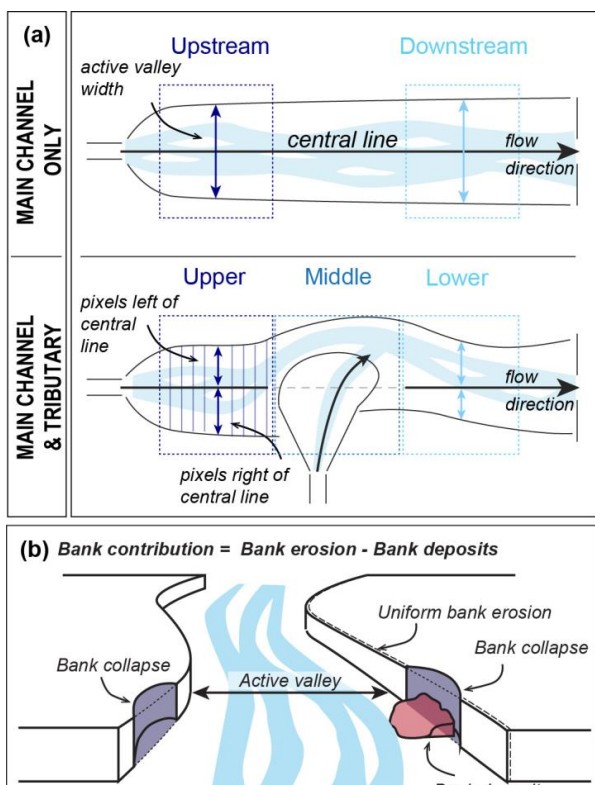


Figure 3. (a) Schematic representation of the method used to calculate the active valley width
and symmetry. Symmetry and averaged width values are calculated for 700 cross sections
located within the boxes marked in the upper panel. The averaged position of the valley margins
with respect to an imaginary central line, which connects the source zone to the outlet of the
wooden box, is shown in Figure 7. This representation highlights the symmetry of the valley and
indirectly provides the valley width (i.e., sum of the right and left-margin positions). Boxes
marked in the lower panel show the division in Upper, Middle, and Lower sections used for the
calculation of the mobilized volumes (Fig. 9). (b) Schematic representation of the method used to
calculate bank contribution: Elevation difference > -2.5 cm represents bank erosion and bank
collapses, whereas differences > 2.5 cm represent large bank deposits. The contribution of the
banks is calculated by subtracting these two values.



### 3.2.3. Sediment discharge at the outlet ($Q_{s\_out}$), mobilized volumes, and bank contribution

The sediment discharge at the outlet of the basin ($Q_{s\_out}$) was manually recorded at 10-minute intervals by measuring the volume of sediment that was collected in a container over a 10-second period. $Q_{s\_out}$ was also calculated by differencing subsequent DEMs (generating a "DEM of Difference", or DoD) and calculating the net change in sediment volume within the DEM. Although having a lower temporal resolution than the manual measurements (i.e., DoDs are averaged over 30 minutes), this DEM-based calculation allowed us to identify zones of aggradation and incision within the system and calculate their volumes. For each DoD, we distinguished between changes along the active valley floor due to channel dynamics (elevation difference < 2.5 cm, value chosen as best cut-off value) and changes that occur along the channel and valley walls, for example due to bank collapses (elevation difference > 2.5 cm). Changes within the active valley floor were further divided into areas of net *aggradation* ($\Delta V_{vf} > 0$) and *erosion* ($\Delta V_{vf} < 0$). Changes in bank elevation were divided into *bank deposition* ($\Delta V_b > 0$) and *bank collapses* or *erosion* ($\Delta V_b < 0$). These were used to calculate the bank contribution ($V_b$) to the total volume ($V$) of mobilized sediment (Fig. 3b). We separated the upper, middle, and lower sections of the experimental river valley by dividing the DEMs into three different zones (Fig. 3a, lower panel). For each section, we calculated the volume of sediment moved between two time steps within the active valley floor ($V_{vf}$), along the banks ($V_b$), and the sum of the two contributions ($V = V_{vf} + V_b$).

The volumes are normalized to the $Q_{s\_in}$ measured over 30 minutes (to match the 30-minute period of a DoD). Negative $V$ values indicate net incision, whereas positive values indicate net aggradation. $V$ values close to zero may indicate that there was no change, or that the net incision $\cong$ net aggradation. As such, it is important to look at the variations through time rather than at single values.



# 4. Results

All experiments included an initial adjustment phase characterized by high $Q_{s\_out}$ and a short and rapid increase in the main-channel slope through preferential channel incision at the downstream end of the main channel. This phase represents the adjustment from the manually constructed valley shape to the shape that is equilibrated to the imposed boundary conditions. At the start of the adjustment phase, the channel rapidly incised toward the outlet, which was much lower than the height of the manually constructed valley bottom, meanwhile depositing material at the channel head adjusting to the $Q_{s\_in}$ and $Q_w$ values. Analogous to a base-level fall observed in nature, this caused an increase in main-channel slope near the outlet and the upstream migration of a diffuse knick-zone that lowered the elevation of the main channel. After this initial adjustment, which marks the end of the spin-up phase, the main controlling factors for the shape of the channel were the $Q_{s\_in}$ and $Q_w$ values only.

## 4.1. Geometric adjustments

Channel-slope adjustments in our experiments followed the theoretical models described above (Section 2.1). Following the spin-up phase, the main-channel slope decreased in all experiments through incision at the upstream end, except for T_NC2 and the initial phase of T_IWMC, in which the boundary conditions favored aggradation (Fig. 4, Table 1). The slope of the tributary increased during periods of fan aggradation (e.g., IS phase of the T_ISDS run, and DW phase of the T_DWIW run) and decreased during periods of fan incision (DS phase of the T_ISDS run, and IW phase of the T_DWIW run) (Fig. 4). Slope adjustments did not occur uniformly, but followed a top-down or bottom-up direction depending on the origin of the perturbation (e.g., changes in headwater conditions or base-level fall at the tributary outlet).



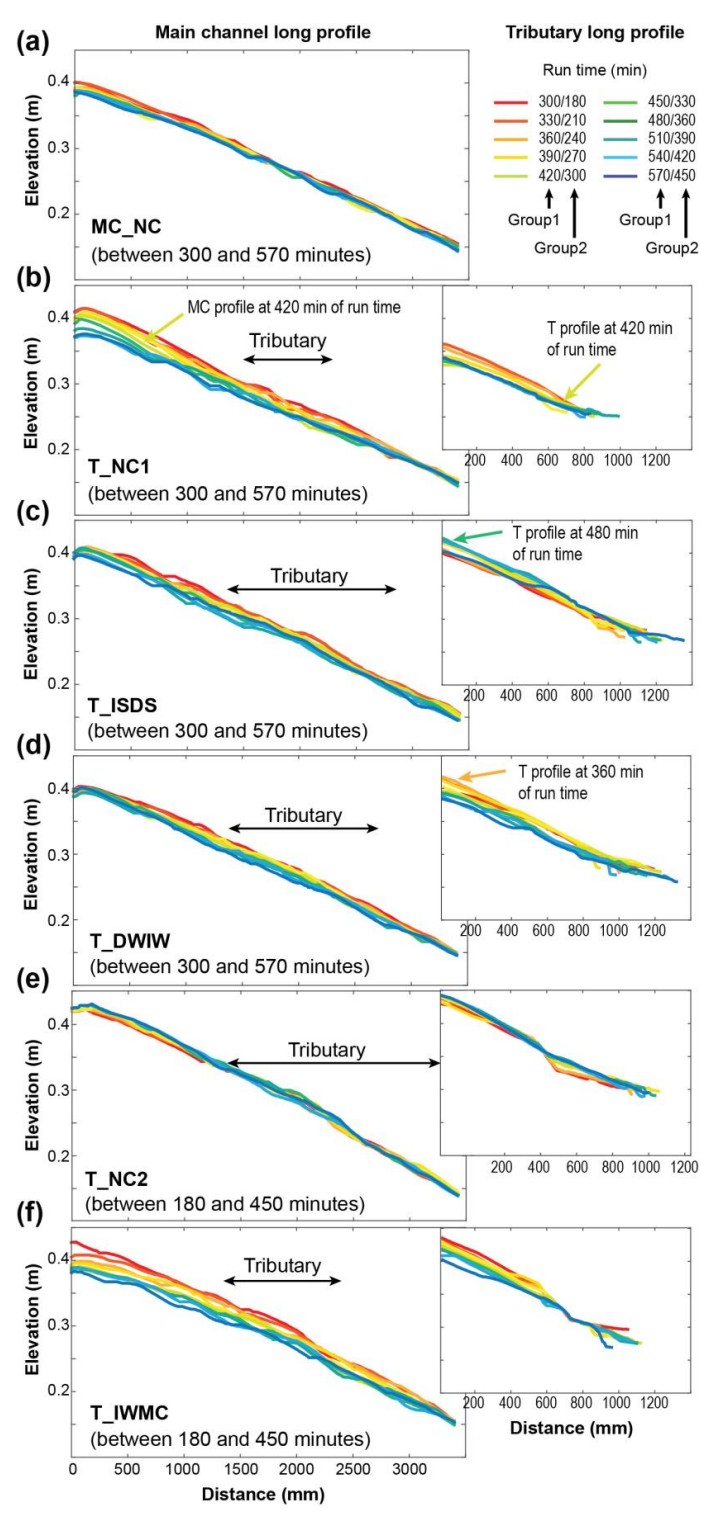





Figure 4. Long profiles of the main channel (left panels) and of the tributary channel (right panels) for all runs. Profiles represent the experiments between 300 and 570 minutes for the MC_Ctrl2, T_NC1, T_ISDS, and T_DWIW runs (legend values to the left of the slashes), and between 180 and 450 minutes for the T_NC2, and T_IWMC runs (legend values to the right of the slashes). Along the main channel profiles, horizontal arrows indicate the position and extent of the tributary channel/alluvial fan, whereas colored arrows indicate the position of the channels in particular run times discussed in the text.

Valley width in both the main channel (Fig. 5) and the tributary (Fig. 6) increased during the experiments, mainly through bank erosion and bank collapses, until reaching relatively steady values (Fig. 7). The experiments with the tributary (Fig. 7b – f) developed a much wider main-channel valley, especially downstream of the tributary, where $Q_w$ was increased > 60% by the additional $Q_w$ input from the tributary. In these experiments, valleys were also strongly asymmetrical, with more erosion affecting the valley side opposite the tributary (Figs. 5 and 7).



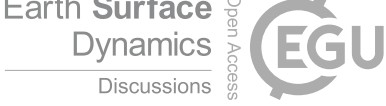

Figure 5. Left panels: Cross sections obtained from the DEMs at three different locations along the main channel (p1, p2, and p3 respectively). The color code represents successive DEMs as illustrated in Fig. 4 (i.e., same colors for the same run times). All cross sections are drawn from left to right looking in the downstream direction. Right panels: DEM maps expressed in meters; color code represents the elevation with respect to the channel floor (also in meters).

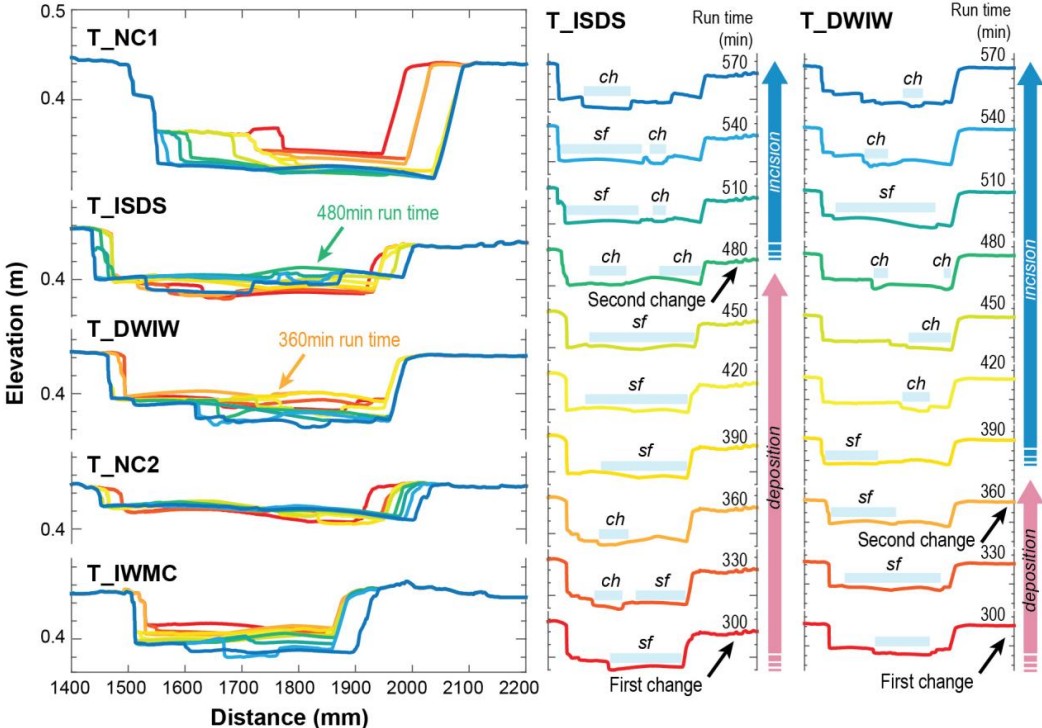

Figure 6. Cross sections in the tributary drawn from left to right looking downstream. The left panels show the evolution of all runs (color code as in Fig. 4 and 5); the right panels show the evolution of the T_ISDS and T_DWIW runs in more detail: the ground-surface elevation (colored lines) and the wetted areas (light blue bars) are shown. During aggradation, sheet flows (*sf*) dominate the transport mode of sediment, although channels (*ch*) may contemporaneously be present on the fan surface. During incision, the flow alternates between channelized flows and sheet flows and contribute to lowering the entire fan topography.









Figure 7. Variations in the geometry of the active valley floor for all experiments. For each
experiment the upper panel shows the measured slope (measured every 10 minutes during each
experimental run). The middle panel shows the calculated average position of the right and left
valley margins with respect to the central line, respectively for the main channel upstream and
downstream of the tributary junction (as indicated in Fig. 3a). Gray areas represent the spin-up
phase of each experiment (based on the break-in-slope registered through the manual slope
measurements; (a–f) upper panels). Vertical dotted lines in the T_ISDS, T_DWIW, and
T_IWMC runs represent the *time of change* in boundary conditions. Values are reported with
their relative 1σ value. For all experiments with a tributary, the shape of the fan and the dominant
sedimentary regime acting in the tributary at that specific time (i.e., vertical incision (VI), lateral
erosion (LE), or aggradation (A)) is shown in the lower panel. In all experiments, fan-toe cutting
(Leeder and Mack, 2001; Larson et al., 2015) mainly occurred at the upstream margin of the fan
and contributed to the strong asymmetry of the fan morphology (Table S9 of Supp. Material),
similar to what has been observed in nature (Giles et al., 2016).

### 4.2. $Q_{s\_out}$ and bank contribution
Our experiments offered a rare opportunity to evaluate the impacts of sediment supply from
the tributary to the main channel through space and time. In general, sediment moved in pulses,
and areas of deposition and incision commonly coexisted (Fig. 8a).
$Q_{s\_out}$ varied greatly, but generally decreased through time (the only exception is the
T_IWMC run, where $Q_{s\_out}$ remained high) (Fig. 8, black circles). Values for the mobilized
sediment, *V,* calculated from the DoDs (averaged over 30 minutes) show similar trends, but with
a lower variability that reflects the long-term average $Q_{s\_out}$ (Fig. 8, black lines). An appreciable
reduction of $Q_{s\_out}$ occurred when the system was approaching equilibrium (e.g., end of Fig. 8a,
b) and during times of fan aggradation in the tributary (i.e., IS and DW phases of Fig. 8c, d, and
e). Net mobilized sediment volumes (*V*) increased again during phases of fan incision (i.e., DS
and IW phases of Fig. 8c and d) and main-channel incision (e.g., IW phase in Fig. 8f). These
increases were due to the combined effect of a general increase in sediment mobility within the
active valley floor ($V_{vf}$) and lateral erosion of the banks ($V_b$) (Fig. 8, violet and orange bars
respectively, and Fig. S6 of the Supp. Material). The DoD analysis also indicates that in all
experiments, with the only exception of the MC run and of the phases approaching steady-state,
bank contribution was higher or of the same order of magnitude of the volume mobilized in the
valley floor (Fig. 8, orange and violet bars). This suggests that bank erosion represented a major
contribution to $Q_{s\_out}$ (Tables S3 to S8 of Supp. Material). This is particularly true also for the





Earth **Surface**
**Dynamics**
Discussions

T_NC2 run, where aggradation was favored, in which $Q_{s\_out}$ is dominated by the contribution of
the banks (Fig. 8e, and Fig. S7 of the Supp. Material).

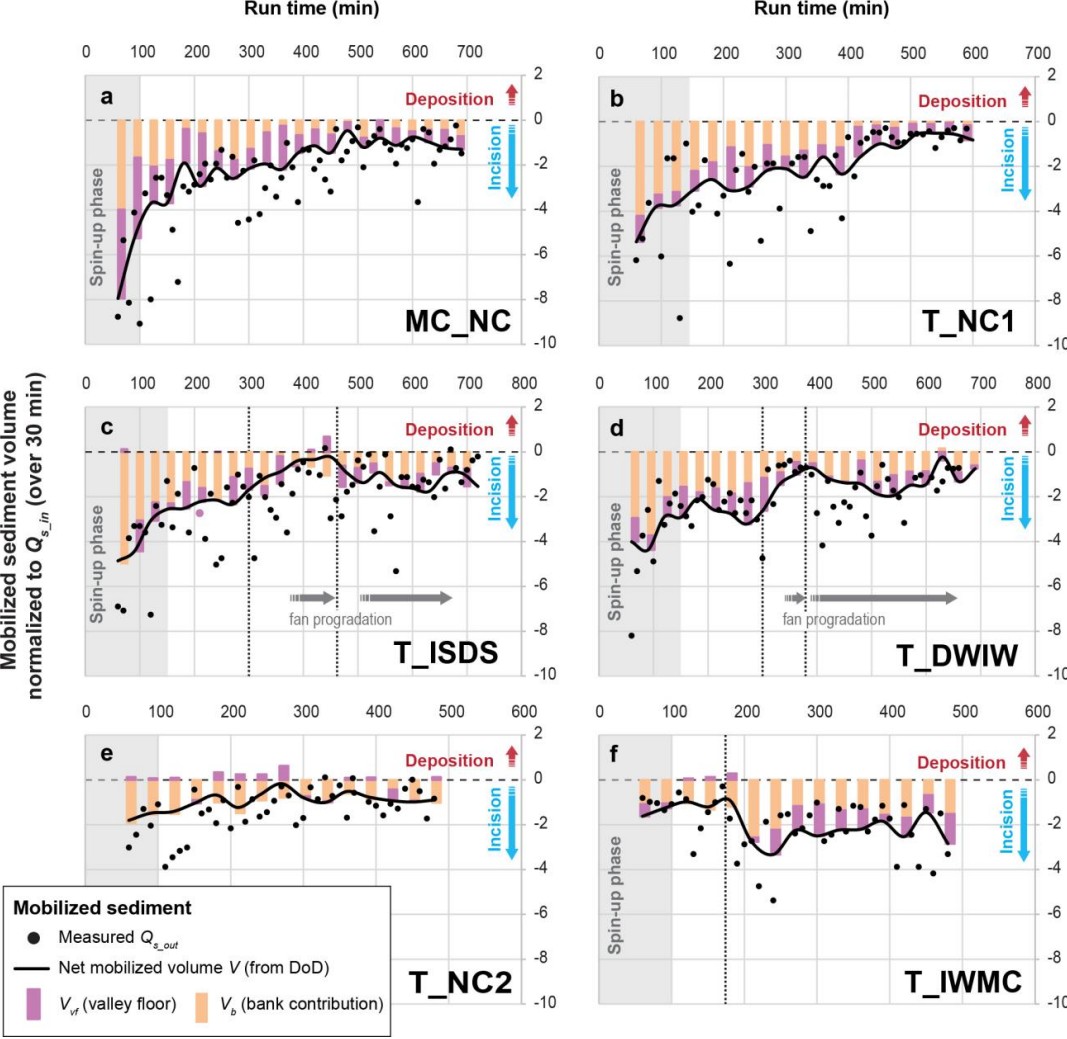

Figure 8. Volumes of sediment mobilized within the system. Black line: Net mobilized volume
of sediment measured using the DoD. For comparison, black dots represent the $Q_{s\_out}$ values
measured every 10 minutes (part of the difference between measured and calculated $Q_{s\_out}$ values
may be due to the contribution of the most downstream area of the wooden box, which was
shielded in the DEM reconstruction). Horizontal arrows indicate the timespan of fan
progradation either during fan aggradation or fan incision. Vertical pointed lines represent the
*time of change* in boundary conditions; horizontal dashed line separates aggradation and erosion.




### 4.3. Downstream sediment propagation

To analyze the effects of the tributary on the mobility of sediment within the coupled tributary–main-channel system, we monitored the volumes of sediment mobilized ($V$) in the upper, middle, and lower sections of the fluvial network through time (Fig. 9). The complex pattern of $V$ in the different sections yields insights into downstream sediment propagation, especially when coupled with maps of the spatial distribution of eroded and deposited sediment (Fig. 10, and Figs. S1 to S5 in the Supp. Material):

1. In all experiments, including the one without a tributary (MC_NC), sediment moved in pulses through the system (Fig. 9). As such, the mobilized volumes ($V$) of each section can be *in-phase* or *out-of-phase* with the volumes mobilized in the others sections (Castelltort and Van Den Driessche, 2003) depending on where the "pulse" of sediment was located within the floodplain (Fig. 11a).

2. The sediment mobilized in the middle and lower sections of the T_NC1 run showed a decrease in $V$ after ca. 400 min, whereas in the upper section $V$ remained nearly constant (Fig. 9b), despite a marked increase in $V_{vf}$ (Fig. S6 of Supp. Material).

3. In the T_ISDS run, the middle section showed, as expected, a strong reduction in $V$ after the onset of increased $Q_{s\_in}$ in the tributary and consequent fan aggradation (300 to 480 minutes). Conversely, it showed an increase in $V$ following the decrease in $Q_{s\_in}$ and consequent fan incision (480 minutes to the end of the run) (Fig. 9c). A similar pattern can be seen in the lower section, with a reduction in $V$ during fan aggradation and an increase in $V$ during fan incision. Interestingly, the upper section showed two peaks of enhanced $V$ (i.e., increase in sediment export) just after the changes in the tributary, followed by a prolonged reduction of $V$ (i.e., decrease in sediment export) during phases of fan progradation.

4. Patterns similar to those described for the T_ISDS can be seen for the T_DWIW run. However, due to the type of change in the tributary (i.e., decrease in $Q_w$, which increases the $Q_s/Q_w$ ratio, reducing the sediment-transport capacity) and due to the shorter duration of the perturbation (300 to 375 minutes), the first peak of enhanced $V$ in the upper section was barely visible, whereas the second peak was not present. Rather, the upper



section shows a continuous decrease in $V$ until ca. 420 min, i.e., circa 45 minutes after
the onset of increased $Q_w$ in the tributary (Fig. 9d and Fig. S3 of Supp. Material).
5.   The T_NC2 experiment is dominated by aggradation and $V$ values are rather constant;
(Fig. 9e and Fig. S4 of Supp. Material). Similar to the final part of the T_NC1 run, the
upper section of the main channel showed a general increasing trend in $V_{vf}$ (Fig. S7 of
Supp. Material).
6.   In the T_IWMC experiment, as expected, $V$ increased immediately after the increase in
$Q_w$ in main channel in all three sections (indicating major incision), but was particularly
evident in the upper and lower sections of the main channel (Fig. 9f).

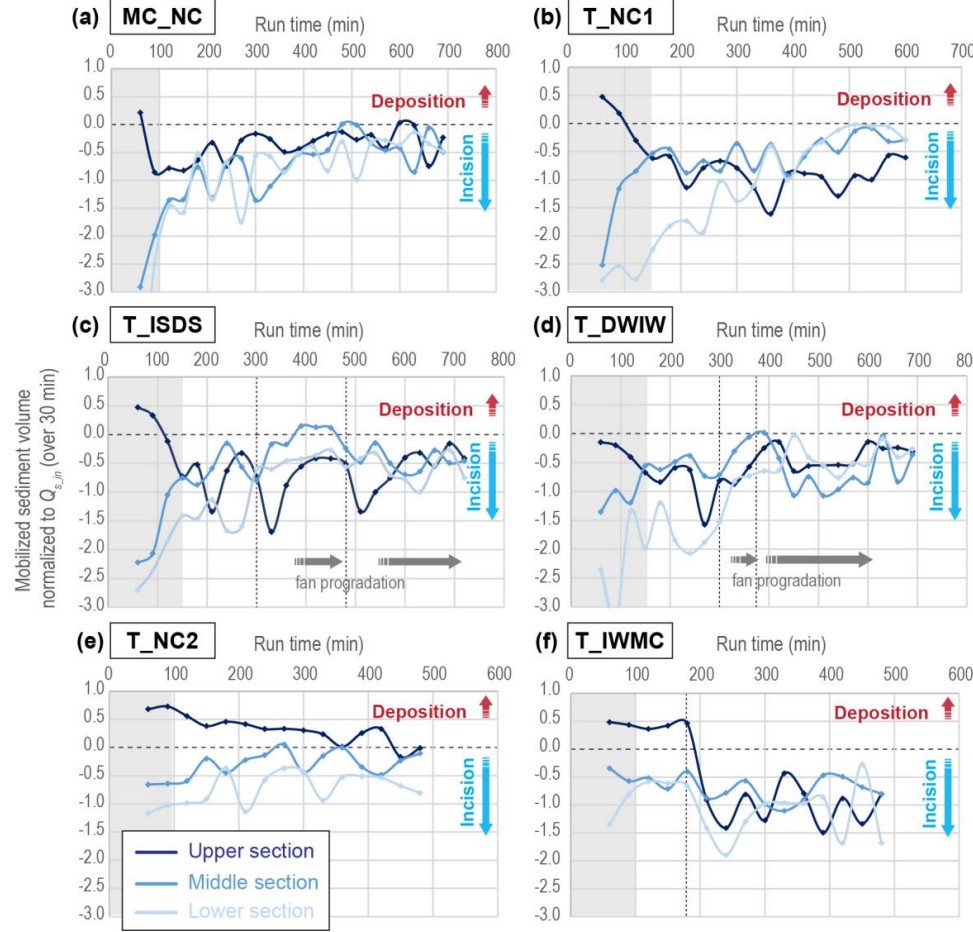




Figure 9. Volume (*V*) of sediment mobilized in each section (e.g., upper, middle, and lower
sections). Vertical lines represent the *times of change* in boundary conditions; horizontal dashed
line separates aggradation and erosion.

Earth **Surface**
**Dynamics**
Discussions


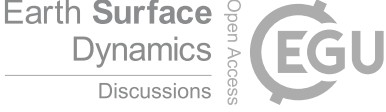

Figure 10. Sediment transfer dynamics within the system in the T_ISDS experiment (from DoDs
analysis). Variations between -0.001 and +0.001 m are considered as "no change" (in gray) to
account for the DEMs accuracy (i.e., 1 mm resolution). (a) Pre-perturbation phase (between 30
and 150 minutes is considered to be the spin-up phase); (b) Fan aggradation (300-390 min) and
progradation (390-480 min) phase; (c) Fan incision and progradation phase (480 min until end of
run).

## 5. Discussion

Our six experiments provide a conceptual framework for better understanding how tributaries
interact with main channels under different environmental forcing conditions (Fig. 1). We
particularly considered geometric variations of the two subsystems (i.e., tributaries and main
channels) and the effects of tributaries on the downstream delivery of sediment within the fluvial
system.

### 5.1. Aggrading and incising fans: geometrical adjustments and tributary–main-channel interactions

In our experiments, the aggrading alluvial fans strongly impacted the width of the main-
channel valley both upstream and downstream of the tributary junction. By forcing the main
channel to flow against the valley-wall opposite the tributary, bank erosion was enhanced, thus
widening the main-channel valley floor (Figs. 4, 7, and 10). Bank erosion and valley widening in
the main channel also occurred during periods of fan incision (Figs. 10b, S3, and S6 of the Supp.
Material). We hypothesize that this widening was related to pulses of sediment eroded from the
fan, which periodically increased the sediment load to the main channel and helped to push the
river to the side opposite the tributary (Grimaud et al., 2017; Leeder and Mack, 2001). Once
there, the river undercut the banks, causing instability and collapse. As such, periods of fan
incision triggered a positive feedback between increased load in the main channel and valley
widening, which occurred mainly through bank erosion and bank collapses. In these scenarios,
bank contribution ($V_b$) in the middle and lower sections of the main channel can be equal to, or
larger than, the sediment mobilized within the active valley floor ($V_{vf}$) (also for the T_NC2 run;
Fig. 8b and Fig. S6 and S7, Supp. Material). It follows that the composition of the fluvial
sediment may be largely dominated by material mobilized from the valley walls, with important
consequences, for example, for geochemical or provenance studies (Belmont et al., 2011).



Our analysis of sediment mobility within the different sections of the main channel
highlighted that the presence of the alluvial fan affects the time needed to reach equilibrium in
the different reaches of the main river: in the T_NC1 run, for example, due to the sediment input
from the tributary, the middle and lower sections have a higher $Q_s/Q_w$ ratio (0.022) than the
upper section (0.014), and may reach equilibrium faster (Gilbert, 1877; Wickert and Schildgen,
2019). Once the tributary reached equilibrium (e.g., at ca. 420 minutes for T_NC1; inset of Fig.
4b), the upper main channel rapidly adjusted by decreasing the elevation of its channel bed (Fig.
4b) and increasing the sediment mobilized (Fig. 9b and Fig. S6 of Supp. Material). This result
suggests that equilibrium time scales of channels upstream and downstream of tributaries can
vary (Schumm, 1973), and that in a top-down direction of adjustments, the equilibrium state of
the upper section may be dictated by the equilibrium state of its lower reaches because of the
tributary influence.
In our experiments, fans were built under conditions that caused deposition at the tributary
junction (e.g., an increase in $Q_{s\_in}$ or decrease in $Q_w$ in the tributary). When the perturbation
lasted long enough (e.g. in experiment T_ISDS), the fan prograded into the main channel. The
passage from fan aggradation to progradation was delayed relative to the onset of the
perturbation by the time necessary to move the sediment from the fan head to the fan margin
(e.g. for > 60 min in T_ISDS; Fig. 10b). This delay allowed for a temporarily efficient transfer of
sediment within the main channel (as marked by the peak in $V$ of the upper main channel section;
Fig. 9c). For tributaries subject to a change that caused tributary incision (e.g., decrease in $Q_{s\_in}$
or increase in $Q_w$), the elevation of the fan surface was progressively lowered (inset of Fig. 4c
and d, and Fig. 6), and the fan prograded into the main channel with cyclic pulses of sediment
discharge (e.g., Fig. 10c) (Kim and Jerolmack, 2008). Progradation was generally localized
where the tributary channel debouched into the main river (e.g., depositing the *healing wedge* of
Leeder and Mack, 2001), generally shortly after (< 30 min) the onset of the perturbation (Figs.
10c and S3 of the Supp. Material). When the fan prograded, sediment in the main channel was
blocked above the tributary junction (e.g., at 390 to 480 min in Fig 10b, and at 510 min to the
end of the run in Fig.10c; Fig. S4 of Supp. Material), and the upstream main-channel section
experienced a prolonged decrease in sediment mobility due to localized aggradation (Fig. 9c and
d, and Fig. 11b).



Given the relative size of the tributary and main channel in our experiments ($Q_w$ tributary ~
2/3 $Q_w$ main channel) and the magnitude of the perturbations (doubling of $Q_{s\_in}$ or halving of
$Q_w$), the impact of perturbations in the tributary on the sediment mobility ($V$) within the main
channel remained mostly within autogenic variability (Fig. 8b, Group 1). This observation
highlights how the analysis of changes in $Q_{s\_out}$ alone (for example inferred from the stratigraphy
of a fluvial deposit) may not directly reflect changes that occurred in the tributary, but can be
overprinted by autogenic variability. However, the analysis of $V$ within individual sections of the
main channel, and particularly within the confluence zone (i.e., middle section), together with the
analysis of how sediment moves in space, reveal important changes in the sediment dynamics of
the main channel that may help to reconstruct the perturbations that affected the tributary
(Section 5.2; Figs. 9 and 11b). This observation underscores the need to study both the tributary
and main-channel deposits (Mather et al., 2017), both upstream and downstream of a tributary
junction.



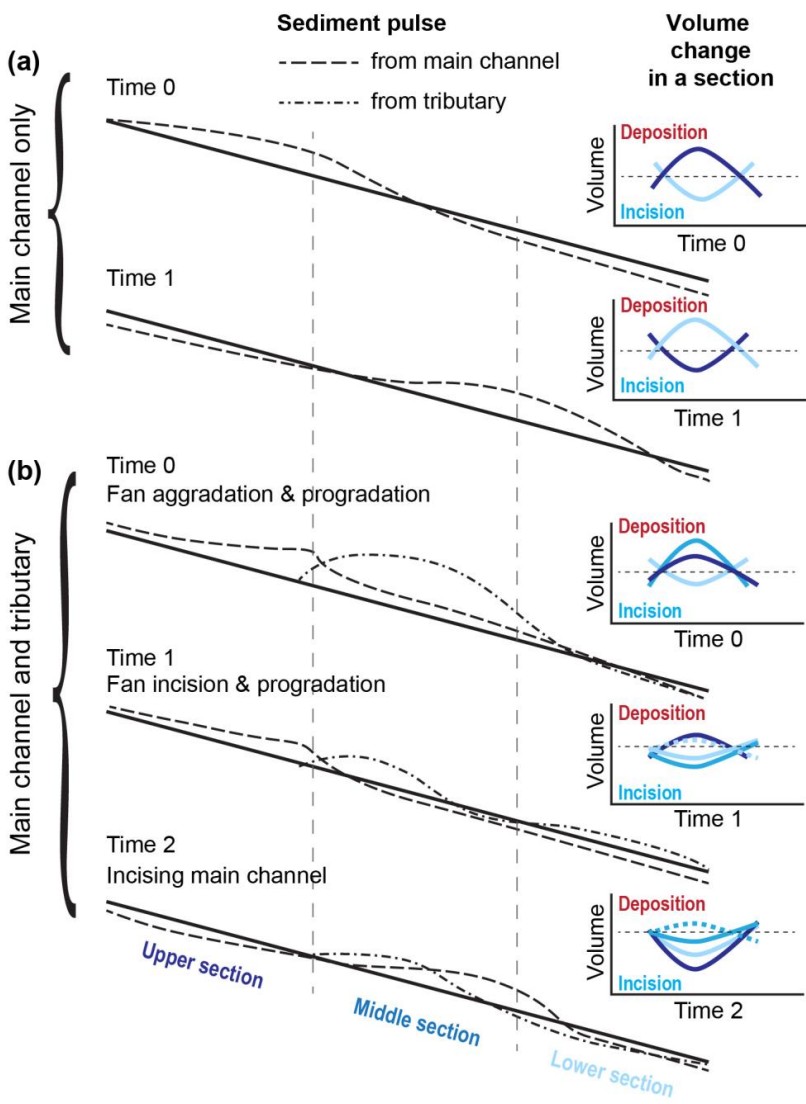


Figure 11. Schematic representation of the average sediment mobilized in each section of the
main channel. Solid black line represents the idealized equilibrium profile of the main channel,
whereas dashed lines represent the volumes mobilized from the main channel and from the
tributary. (a) Sediment dynamics in a single-channel system: sediment moves in pulses and upper
and lower sections may be *out-of-phase* or *in-phase* depending on the dynamics of the middle
section (i.e., the *transfer zone* of Castelltort and Van Den Driessche, 2003). (b) Sediment
dynamics in a tributary-main channel system: *Time 0* represents the "aggrading (and prograding)
fan" scenario, where the upper and middle sections of the main channel undergo aggradation,
while the lower section undergoes incision. *Time 1* represents the "incising (and prograding) fan"
scenario, where the upper section may still be aggrading by it also starts to get incise creating a
pulse of sediment that reaches the lower section. The middle section clearly sees an increase in





incision due to the imposed perturbation, while the lower section may undergo incision or
aggradation depending on the amount of sediment delivered from the fan, from the upper section,
and from bank erosion. *Time 2* represents the "incising main channel" scenario, where the fan
loses its influence on the dynamics of the main channel and both upper and lower sections
undergo incision. The middle section can undergo aggradation or incision depending on the
amount of sediment mobilized in the tributary and on the pulse of sediment moving from the
upper to the lower section of the main channel.

5.2. Incising main channel: geometric adjustments and tributary–main-channel

interactions

The main-channel bed elevation dictates the local base level of the tributary, such that
variations in the main-channel long profile may cause aggradation or incision in the tributary
(Cohen and Brierly, 2000; Leeder and Mack, 2001; Mather et al., 2017). In our experiments,
lowering of the main-channel bed triggered tributary incision that started at the fan toe and
propagated upstream (insets in Fig. 4). Because tributary incision increases the volume of
sediment supplied to the main channel, a phase of fan progradation would be expected, similar to
the cases described above (and in the *complex response* of Schumm, 1973). However, in our
experiment (i.e., T_IWMC), progradation did not occur: instead, the fan was shortened (Fig. S5
Supp. Material). We hypothesize that the increased transport capacity of the main river resulted
in an efficient removal of the additional sediment from the tributary, thereby mitigating the
impact of the increased sediment load supplied by the tributary to the main channel. Another
consequence is that the healing wedge of sediment from the tributary is likely not preserved in
the deposits of either the fan margin or the confluence zone, hindering the possibility to
reconstruct the changes affecting the tributary. However, some insight can be obtained from the
analysis of sediment mobility. During main-channel incision, whereas both upper and lower
sections of the main channel registered a marked increase in *V* following the perturbation, the
middle section showed only minor variations (Fig. 9f). We hypothesize that this lower variability
was due to the buffering effect of the increased load supplied from the fan undergoing incision
(i.e., caused by the sudden base-level fall that followed main-channel incision) (Fig. 11b). In
contrast, when incision in the tributary was caused by a perturbation in its headwaters, *V* initially
increased and then showed a prolonged decrease in the upper section during fan aggradation,
whereas it increased in the middle section during fan incision. These differences may help to



discern the cause of fan incision (i.e., either a perturbation in the main channel or in the
tributary).
We did not observe the *complex response* described by Schumm (1973), characterized by
tributary aggradation following incision along the main channel. That complex response likely
occurred because the main river had insufficient power to remove the sediment supplied by the
tributaries, as opposed to what occurred in our experiments. When aggradation occurs at the
tributary junction, one may expect to temporarily see an evolution similar to that proposed in the
"aggrading alluvial fan" scenario, with the development on an alluvial fan that may alter the
sediment dynamics of the main channel, modulating the sediment mobilized in the upper and
lower sections of the river and delaying main-channel adjustments. In our experiment, instead, a
prolonged erosional regime within the main channel may have led to fan entrenchment and fan-
surface abandonment (Clarke et al., 2008; Nicholas and Quine, 2007; Pepin et al., 2010; Van
Dijk et al., 2012). Despite the lack of fan progradation, an increase in bank contribution
following incision of the main channel did occur (Fig. 8b.6, Fig. S7 Supp. Material) and could be
explained by (1) higher and more unstable banks and (2) an increased capacity of the main
channel to laterally rework sediment volumes under higher water discharges (Bufe et al., 2019).
5.3. Sediment propagation and coupling conditions
Understanding the interactions between tributaries and main channel, and the contribution of
these two sub-system to the sediment moved (either eroded or deposited) in the fluvial system, is
extremely important for a correct interpretation of fluvial deposits (e.g., cut-and-fill terraces or
alluvial fans), which are often used to reconstruct the climatic or tectonic history of a certain
region (e.g., Armitage et al., 2011; Densmore et al., 2007; Rohais et al., 2012).
In their conceptual model, Mather et al. (2017) indicate that an alluvial fan may act as a
*buffer* for sediment derived from hillslopes during times of fan aggradation, and as a *coupler*
during times of fan incision, thereby allowing the tributary's sedimentary signals to be
transmitted to the main channel. From our experiments, we can explore the effects that tributaries
have not only in storing or releasing sediment to the main channel, but also in modulating the
flux of sediment within the fluvial system. In doing so, we create a new conceptual framework
that takes into account the connectivity within a coupled alluvial fan-main channel system and





the mechanisms with which sediment and sedimentary signals may be recorded in local deposits
(Fig. 12). Results are summarized as follows:

*5.3.1.  Aggrading and incising fans*

1.  If the tributary has perennial water discharge, a *partial coupling* between the tributary

and the main channel is possible. Also, during fan aggradation, when most of the

sediment is deposited and stored within the fan (e.g., Fig 10b), a portion of the $Q_{s\_in}$

reaches the main channel in proportion to the transport capacity of the tributary channel

(Fig. 12a and b). The partial coupling between the fan and the main channel allows for a

*complete coupling* between the upstream and downstream sections of the main river (Fig.

10b – 300-390 min, and S3b in the Supp. Material). As such, during fan aggradation, the

main channel behaves as a single connected segment, and the lower section receives

sediment in proportion to the transport capacity of the main and tributary channels. The

material supplied by the tributary to the main channel is dominated by the tributary's

$Q_{s\_in}$ with little remobilization of previously deposited material.

2.  During fan incision, large volumes of sediment are eroded from the fan and transported

into the main channel as healing wedges, allowing the fan to progade into the main

channel (Fig. 10c and 12c). This process creates a *complete coupling* between the

tributary and the main channel (Fig. 9c and d), with the material supplied by the tributary

mostly dominated by sediment previously deposited within the fan.

3.  During times of fan progradation, the fan creates an obstacle to the transfer of sediment

down the main channel, creating a *partial decoupling* between upstream and downstream

sections of the main channel (Fig. 9, 10b and c, and 12b and c). As a consequence, the

sediment carried by the main channel is trapped above the tributary junction and thus will

be missing from downstream sedimentary deposits. However, the upstream section of the

main channel may be periodically subject to incision (e.g., Fig. 10b and c), moving

mobilized sediment from the upper to the lower section. Accordingly, if progradation of

the fan is due to prolonged fan aggradation, the downstream section will receive the $Q_{s\_in}$

from the fan, plus pulses of sediment eroded from the upstream section of the main

channel. Conversely, if progradation is due to incision of the tributary and mobilization

of additional fan sediment, the downstream section will receive pulses of erosion from



either the fan or the upstream section of the main channel, plus the contribution of bank

erosion.

In summary, downstream fluvial deposits record the competition between the main

channel and the tributary: the alluvial fan pushes the main channel towards the opposite side

of the valley to adjust its length, whereas the main channel tries to maintain a straight course

by removing the material deposited from the fan. If the main channel dominates, it cuts the

fan toe and permits sediment from upstream of the junction to be more easily moved

downstream. If the tributary dominates, the main channel will be displaced and the transfer of

sediment through the junction will be disrupted. An autogenic alternation of these two

situations is possible, whereby fan-toe cutting may trigger fan incision and progradation,

increasing the influence of the fan on the main channel. The composition of the sediment

downstream thus reflects the competition between main channel and alluvial fan, with

contributions from both sub-catchments. In addition, bank erosion may make important

contributions to sediment supply and transport, particularly during periods of fan incision

(Fig. S6 in the Supp. Material). From these results, we therefore distinguish between: 1)

*Influential alluvial fans*, which have a strong impact on the geometry and sediment-transfer

dynamics of the main channel, and 2) *Non-influential alluvial fans*, which do not

substantially alter the geometry or sediment-transfer dynamics of the main channel.





Figure 12. Conceptual framework for the coupling conditions of an alluvial-fan/main-channel (*MC*) system under different environmental forcings. For *aggrading and incising alluvial fans* (upper panels), the fan-main channel connectivity depends on the dynamics acting in the alluvial fan, being partially coupled during fan aggradation and totally coupled during fan incision. For *incising main rivers* (lower panel) the fan and main channel are fully coupled. As well, *non-influential alluvial fans* (left and lower panels) favors a complete coupling within the main channel, whereas *influential alluvial fans* (middle and right upper panels) may favor a partial decoupling between upstream and downstream sections of the main river. Each one of the four settings presented here brings its own sedimentary signature, different responses to perturbations, and dynamics of signal propagation which may be recorded into the fluvial deposits.

### 5.3.2. Incising main channel

1. Lowering of the main-channel bed triggers incision into the alluvial fan, thereby promoting a *complete coupling* between the fan and the main channel (Fig. 12d, and S5 in the Supp. Material). The sediment supplied by the tributary is mainly composed of material previously deposited within the fan.

2. An increase in main-channel water discharge increases the transport capacity of the mainstem so that it persistently "wins" the competition with the alluvial fan. In this case, despite the incision triggered in the alluvial fan, which increases the sediment supplied by the tributary, the main channel efficiently removes the additional sediment load, thereby reducing the influence of the alluvial fan on downstream sediment transport within the main channel (Fig.S5 in the Supp. Material). The consequence is a *complete coupling* between the upstream and downstream sections of the main channel (Fig. 12d). The sediment reaching the lower section is a mixture of eroded material from the main channel, within the fan, and along the banks.

## 6. Conclusion

We performed six experiments to analyze the interactions of a tributary–main-channel system when a tributary produces an alluvial fan. We found that differing degrees of coupling may be responsible for substantial changes in the geometry of the main channel and the sediment transfer dynamics of the system. In general, we found that the channel geometry (i.e., channel slope and valley width) adjusts to changes in sediment and water discharge in accordance with theoretical models (e.g., Ferguson and Hoey, 2008; Parker et al., 1998; Whipple et al., 1998; Wickert and Schildgen, 2019). Additionally, by analyzing the effects of the tributary-main



channel interactions on the downstream delivery of sediment, we have shown that the fluvial

deposits within the main channel above and below the tributary junction may record

perturbations to the environmental conditions that govern the fluvial system.

Our main results can be summarized as follows (Fig. 12):

(1) Fan aggradation leads to a partial coupling between the fan and the main channel, which

permits a complete coupling between the main-channel reaches upstream and downstream of the

tributary junction. As such, the provenance of downstream sediment reflects the dynamics of

both sub-catchments (e.g., tributary and main river), and remobilized material from older

deposits will be minimal.

(2) Fan incision favors a complete coupling between the fan and the main channel, and

remobilizes material previously stored in the fan.

(3) Fan progradation (either during prolonged aggradation or fan incision) strongly

influences the main channel. As a result, the connectivity of the main river across the tributary

junction is reduced and the deposits of the fluvial system above and below the junction may

record different processes.

(4) Incision along the main channel triggers incision in the alluvial fan that, despite an

increased sediment supply to the main river, reduces its influence on the dynamics of the main

channel. The result is a fully connected fluvial system in which the deposits record sediment-

transfer dynamics and the interactions between both the alluvial fan and the main river, including

a large component of material remobilized from older deposits.

The theoretical framework proposed in this study aims to illustrate the dynamics acting

within a tributary junction, which is an ubiquitous phenomenon across many environments. It

provides a first-order analysis of how tributaries affect the sediment delivered to the main

channels, and of how sediment is moved through the system under different environmental

forcing conditions. With this information we hope to provide a better understanding of the

composition and architecture of fluvial sedimentary deposits found at confluence zones, which is

essential for a correct reconstruction of the climatic or tectonic histories of a basin.



**Data availability**
Data will be made available.
**Video supplement**
Time-lapse video of the experiment will be uploaded.
**Supplement**
Supplement tables and figures can be found in the supplementary document.
**Author contributions**
SS, ST, and ADW designed and built the experimental setup. SS and ST performed the
experiments. SS analyzed the data with the help of ST, ADW and AB. All authors discussed the
data, designed the manuscript, and commented on it. SS designed the artwork.

**Competing interests**
The authors declare that they have no conflict of interest.

**Acknowledgments**
We thank Ben Erickson, Richard Christopher, Chris Ellis, Jim Mullin, and Eric Steen for
their help in building the experimental setup and installing equipment. We are also thankful to
Jean-Louis Grimaud and Chris Paola for fruitful discussions and suggestions.

**Financial support**
This research has been supported by the Deutsche Forschungsgemeinschaft (grant no. SCHI
1241/1-1 and grant no. SA 3360/2-1), the Alexander von Humboldt-Stiftung (grant no. ITA
1154030 STP), and the University of Minnesota**.**

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
