# Peer review of "Interactions between main-channels and"

_Earth Surface Dynamics, 2019_

## Referee Comment (RC1) · Lucy Clarke (Referee) · 9 Jan 2020

The manuscript presents an experimental investigation into the impact of tributary channels, in particular the presence of alluvial fans, on river channel behaviour. The results from six experiments are presented to understand the impact on channel slope, profile, aggradation/incision patterns and sediment dynamics. This is an interesting study and I believe that it adds to established literature in this field and represents a contribution to scientific knowledge in this area that would be of interest to the readership. I support publication of the manuscript following some modification.

The following aspects should be addressed:

[Figure]

• Section 2 could be reduced and integrated into the general context provided in the first section; there is repetition of much of the material between these sections and an overview of basic theory that could be condensed

• In the methods section there needs to be further clarification on how the input conditions were determined for the experiments, i.e. how were the initial Qw and Qs values decided upon? How was the ratio between tributary and main channel initial size, Qs and Qw calculated? There are different Qw:Qs ratios between group 1 and group 2 to promote aggradation or incision but how did you determine what was an appropriate ratio? Also why was there only one Qw change in the T_IWMC experiment when there were 2 changes for the tributary conditions in the group 1 experiments

• There are a lot of figures in the paper, these are presented to a high quality and are informative but the number is overwhelming at the moment and some consideration could be given to reducing the number of these in the main paper and moving some to the supplementary information (i.e. Figure 6 could be removed, and it is not necessary to have both Figures 9 and 10). Additionally, the figure headings are very long and often repeat what is said in the main text – therefore this information could be removed from one or other of these to make the paper overall more concise.

Minor changes: • Title: suggest revising the word "channel" and being more specific that you are referring to the main/trunk channel in a river • Line 74: there have been some papers that have explored the influence of tributary fans on main channels in the field (i.e. Giles, 2016 that you refer to later) and there should be some description here of what these have shown • Table 1 could be expanded (or a separate table used) to include a brief summary of each of the experiments (this is covered in section 3.2 but a concise summary for reference would be useful) and also including the duration of each experiment. The spin-up time for each could also be stated • Line 352: why is the Qs-out only recorded over a 10 second period rather than over the whole 10 minute recording period? • Line 670 remove the colon at the end of this sentence, or remove the sub-section heading for 5.3.1 and 5.3.2 • Be consistent in your use of

hyphens with certain words, i.e. grain size and grain-size

---

## Referee Comment (RC2) · Luca C Malatesta (Referee) · 11 Jan 2020

Dear Editor,
I have read the new manuscript by Savi and colleagues, Interactions between channels and tributary alluvial fans: channel adjustments and sediment-signal propagation. The authors present the results of six flume experiments where they modelled the dynamics of a tributary stream building a fan onto a trunk channel (both transported-limited with uniform grain size and a discharge ratio 2/3). They tracked the evolution of sediment flux (Qs) and topography after changing water discharge (Qw) or input Qs in either channels. The authors build a classification framework with four cases mapping the

types of interaction between tributary alluvial fans and trunk channels and their likely Qs signature.

The article is well written and the experiments are exhaustively described. While this fluvial configuration is quite particular, it will be a very useful resource for anyone working on similar or related features. The manuscript merits publication in e-surf after some amendments. I have comments related to: 1) the structure or nature of the manuscript as review/experimental paper; 2) potential confusion in parts of the description (text and figure) of the experiments; and 3) technical aspects of the discussion.

I start by general comments on the manuscript and then move to focused remarks before a short list of miscellaneous details.

**Review/experimental paper**

The manuscript tries to strike a balance between review paper and niche flume work which I find uneasy to read. The introduction and the background take up the first 8 pages of the manuscript (more than a quarter of the text). They are well-written and offer a quasi exhaustive, if sometimes repetitive, review of the literature. Besides repeated teasers of the flume work to come, the reader could forget it's an experimental paper until the methods section on page 9. Only then the nitty gritty flume work begins. In my opinion, the readers who are interested in a contribution on such a fairly niche setting will be well versed in most of the concepts detailed in the first pages. One or two refresher paragraphs on the graded stream and the relationships between Qw, Qs, and slope should be enough. Below some examples based from the text.

Section 2
The whole section is a review that I would estimate unnecessary or at least that could be trimmed generously. Only the paragraphs l. 168-172 and l. 224-232 are really

important here because they introduce and contextualize the vocabulary used to describe the experiments.

l. 142-153: this paragraph reads like an introduction and repeats many elements of it. It could be advantageously cut to avoid redundancy.

l. 175-178: this has already been stated and doesn't need to be repeated again.

l. 206-208: reads like an introduction.

l. 239-241: same

If the review should stay, I believe it would be then appropriate to balance the paper and tie up the discussion with reference to the reviewed field sites. It would be particularly strengthening for the framework proposed. For example what would all the one channel studies e.g. Simpson Castelltort be missing by ignoring tributary feedbacks?

**Complex feedbacks as motivation for study**

The potentially important role of tributary feedbacks for buffering or accentuation of environmental signals (l. 63-66, l. 131-132) appears particularly important to me. I would suggest to emphasize it further, and especially to highlight the broader impact to the entire sedimentary system. Maybe you could build a case of how the effects of tributaries could strengthen or weaken the dynamics described by Simpson and Castelltort. That article is well known and I think that it would make your work even more approachable to the reader.

**Motivation for the flume setup.**

Somewhere in the text, maybe in a new section 2, the target landscape of the experiments should be spelled out. The flume seems to be representing the following fluvial landscape: two transport-limited streams (one twice as large as the other) with the same grain size join in a broad alluvial valley/floodplain of unlithified/uncemented sediments. The tributary builds an alluvial fan in the trunk channel. For the case of junctions between alluvial streams of the same order of magnitude Qw and same grain size I would not expect the growth of an alluvial fan. The cases I have in mind where a tributary alluvial fan disturbs a main trunk are higher upstream. Paradigmatic would be the Illgraben Fan growing in Rhône Valley and constraining its river flow. In this case and the many others I can remember, there is an important grain size difference. I think I simply don't have the right references. I suspect that many readers may share the same experience as me. It would therefor be useful to discuss some field sites where the flume setup would apply. Preferably some that were studied for that dynamic.

**Representativity of each model run**

There misses a discussion of the relevance each individual run for the scenario explored. As detailed at length, alluvial systems have rich dynamics with a lot of stochastic processes. How confident are the authors that each run is a representative unique outcome of the scenario tested and not one of a wide range of possible evolutions? I fully understand that this is an inherent limitation of flume studies as each run represents tremendous work but it would strengthen the framework if this limitation is directly addressed in a short paragraph.
**Line by line**

- l. 121-130 The experimental work by Bonnet and Crave (Geology, 2003) on directionality of perturbations in landscapes would be particularly relevant for this paragraph.

- l. 254 It may be good to explicitly write that the level of the water sill is fixed.

- l. 269 I would suggest to point to Table 1 at the end of the first sentence already.

- l. 278-279 This seems a tall order to me. There is a lot of stochastic and non-linear processes in such a system. Wouldn't adding its parts yield more than their sum? Is there a reference for the feasibility of this?

- l. 333-335 This sounds more like the quantification of "straightness" rather than symmetry. The latter implies features within the floodplain to me. maybe add "axial" symmetry? this would make the link with the source-to-outlet straight line clearer.

- l. 367-369 For clarity's sake. V is then the volume of all sediments that were moved in the time interval, regardless whether they exited the section or not. It is the summed volume of all parcels of sediment mobilized during the interval, whether observed as new deposit or as new erosion. However, any sediment bypass would not count toward V regardless of its sediment throughput. I think that this is what I understand from the text.

- l. 381 "deposited"? as in incised and deposited.

- l. 385, l. 389-390: How long is the spin-up phase? Is it 300 minutes after which the changes are observed (Figure 4)? And the spin-up phase is the complete adjustment to boundary conditions, correct?

ESurfD
[Figure]

- l. 546 "mainly" how can the valley widen in other ways than bank erosion?

- l. 557 "once the tributary reached equilibrium": from a slope perspective? It would be useful to restate whether it was after incision or aggradation.

- l. 569-570 Is this change in sediment mobilisation that visible in Qs_out? Or is the lack of tributary Qs merely replaced by main channel Qs during transient phase?

- l. 577-578 "blocked" what is the exact meaning of blocked? Does it mean that 100% of the upstream sediment flux is effectively blocked, or that the sediment flux is limited and part of it is deposited?

- l. 592-593 What kind of deposits are we talking about here? The material buried underneath the floodplain or terrace deposits where available?

- l. 684 one "r" is missing in prograde.

- l. 702-704 The dynamic of that competition must be heavily influenced by the respective erodibility of fan and bank. I imagine that a balanced situation like this one is rare. Tributaries often carry coarser sediment than the floodplain of the main channel. Or conversely floodplain material can be significantly consolidated and much harder to erode than loose fan material. Not even mentioning bedrock-lined valleys. It might be worth discussing comparisons with field examples again here.

- l. 780 how? where?

**Figures**

- Figure 4: This is a very important figure but it is unfortunately hardly readable. Most profiles overlap and any pattern of change is almost impossible to decipher.

Have the authors tried to subtract the elevation along the average slope of the first profile from all profiles? This detrended curve would allow to spread the plots in the vertical. Further, the colour scheme is most likely not colour-blind friendly and should be amended (see Crameri's scientific colour scales for example).

- Figure 7: the small outlines of the fan shapes is a great idea!

- Figure 12: typos in "decoupling". The figure would be much stronger if examples from the field were listed to anchor these cases in a familiar context. What about aggrading main channel? Where does this setting fall?

Good luck to the authors for the revisions,
Best wishes,
Luca Malatesta

---

## Author Comment (AC1) · 19 Feb 2020

**Lucy Clarke (Referee) lclarke@glos.ac.uk**

The manuscript presents an experimental investigation into the impact of tributary channels, in particular the presence of alluvial fans, on river channel behavior. The results from six experiments are presented to understand the impact on channel slope, profile, aggradation/incision patterns and sediment dynamics. This is an interesting study and I believe that it adds to established literature in this field and represents a contribution to scientific knowledge in this area that would be of interest to the reader-ship. I support publication of the manuscript following some modification. The following aspects should be addressed:

> *We thank the reviewer for the support and the constructive review. Answers to the raised points are reported in-line with the review.*

1. Section 2 could be reduced and integrated into the general context provided in the first section; there is repetition of much of the material between these sections and an overview of basic theory that could be condensed

> *As suggested by the reviewer, we strongly reduced Section 2 and moved few of the important information to the introduction (e.g., lines 97-101 in the manuscript version with changes).*

2. In the methods section there needs to be further clarification on how the input conditions were determined for the experiments, i.e. how were the initial Qw and Qs values decided upon? How was the ratio between tributary and main channel initial size, Qs and Qw calculated? There are different Qw:Qs ratios between group 1 and group 2 to promote aggradation or incision but how did you determine what was an appropriate ratio? Also why was there only one Qw change in the T_IWMC experiment when there were 2 changes for the tributary conditions in the group 1 experiments

> *To decide the initial Qw and Qs conditions we calculated the Qw/Qs ratios of ca 40 alluvial rivers of northern Argentina. These ratios ranged between $10^{-2}$ and $10^{-4}$. To define the values for our experiments, we finally ran several (around 10) short test-runs and observed which ratios guaranteed a good balance between sediment transport and deposition. We have added a sentence to explain this choice in the method section.*
>
> *The size of the two channels was defined based on the size of the wooden box. We performed a single change in Qw in the T_IWMC experiment to explore what may happen in a glaciated catchment following the modern rise in temperature and the consequent glacier retreat (similarly to what happened to many mountain rivers). This motivated our choice of a single change in Group 2 compared to Group 1 experiments. We have added a sentence to explain this reasoning in the text.*

3. There are a lot of figures in the paper, these are presented to a high quality and are informative but the number is overwhelming at the moment and some consideration could be given to reducing the number of these in the main paper and moving some to the supplementary information (i.e. Figure 6 could be removed, and it is not necessary to have both Figures 9 and 10). Additionally, the figure headings are very long and often repeat what is said in the main text – therefore this information could be removed from one or other of these to make the paper overall more concise.

> *According to the reviewer's request, we have moved figure 6 and 10 in the supplementary material. We additionally reduced the headings of some figures (Figs. 3, 4, and the new Fig. 6)*

Minor changes:

1. Title: suggest revising the word "channel" and being more specific that you are referring to the main/trunk channel in a river

> *Done.*

2. Line 74: there have been some papers that have explored the influence of tributary fans on main channels in the field (i.e. Giles, 2016 that you refer to later) and there should be some description hereof what these have shown

> *We have added a short description to what Giles et al have described in their work. There are a couple of points in the text that refer to their work (lines 224-228, and 247-248 in the manuscript version with changes).*

3. Table 1 could be expanded (or a separate table used) to include a brief summary of each of the experiments (this is covered in section 3.2, but a concise summary for reference would be useful) and also including the duration of each experiment. The spin-up time for each could also be stated

> *We have added a column with the duration of each experiment and the corresponding spin-up phase to table 1.*

4. Line 352: why is the Qs-out only recorded over a 10 second period rather than over the whole 10minute recording period?

> *Qs_out has been recorded over a 10sec period because the measure has been done manually, with a small container that we used to collect the material exiting the system. A manually measure over the whole 10min period would have been logistically impossible within the experimental set-up.*

5. Line 670 remove the colon at the end of this sentence, or remove the sub-section heading for 5.3.1 and 5.3.2

> *Done.*

6. Be consistent in your use of hyphens with certain words, i.e. grain size and grain-size

> *Thanks for pointing this out. We have checked through the paper and correct the wording.*

---

## Author Comment (AC2) · 19 Feb 2020

**Luca C Malatesta (Referee)** luca.malatesta@unil.ch

Dear Editor, I have read the new manuscript by Savi and colleagues, Interactions between channel sand tributary alluvial fans: channel adjustments and sediment-signal propagation. The authors present the results of six flume experiments where they modelled the dynamics of a tributary stream building a fan onto a trunk channel (both transported-limited with uniform grain size and a discharge ratio 2/3). They tracked the evolution of sediment flux (Qs) and topography after changing water discharge (Qw) or input Qs in either channels. The authors build a classification framework with four cases mapping the types of interaction between tributary alluvial fans and trunk channels and their likely Qs signature. The article is well written and the experiments are exhaustively described. While this fluvial configuration is quite particular, it will be a very useful resource for anyone work-ing on similar or related features. The manuscript merits publication in e-surf after some amendments. I have comments related to: 1) the structure or nature of the manuscript as review/experimental paper; 2) potential confusion in parts of the description (text and figure) of the experiments; and 3) technical aspects of the discussion. I start by general comments on the manuscript and then move to focused remarks before a short list of miscellaneous details.

> *We are thankful to the reviewer for the constructive comments. Our answers and the changes made to the text are reported as in-line comments.*

**Review/experimental paper**

The manuscript tries to strike a balance between review paper and niche flume work which I find uneasy to read. The introduction and the background take up the first 8 pages of the manuscript (more than a quarter of the text). They are well-written and offer a quasi exhaustive, if sometimes repetitive, review of the literature. Besides repeated teasers of the flume work to come, the reader could forget it's an experimental paper until the methods section on page 9. Only then the nitty gritty flume work begins. In my opinion, the readers who are interested in a contribution on such a fairly niche setting will be well versed in most of the concepts detailed in the first pages. One or two refresher paragraphs on the graded stream and the relationships between Qw, Qs, and slope should be enough. Below some examples based from the text.

> *Following the reviewers' comments we have strongly reduced section 2 ('Background') leaving only few background information that may help the reader to better appreciate the results of our study.*

**Section 2**

The whole section is a review that I would estimate unnecessary or at least that could be trimmed generously. Only the paragraphs l. 168-172 and l. 224-232 are really important here because they introduce and contextualize the vocabulary used to describe the experiments.

> *We have moved some of the important lines with the vocabulary in the introductions and strongly reduced the whole section. The following passages, mentioned by the reviewer, have been changed or deleted.*

l. 142-153: this paragraph reads like an introduction and repeats many elements of it. It could be advantageously cut to avoid redundancy.

l. 175-178: this has already been stated and doesn't need to be repeated again.

l. 206-208: reads like an introduction.

l. 239-241: same

If the review should stay, I believe it would be then appropriate to balance the paper and tie up the discussion with reference to the reviewed field sites. It would be particularly strengthening for the framework proposed. For example what would all the one channel studies e.g. Simpson Castelltort be missing by ignoring tributary feedbacks?

**Complex feedbacks as motivation for study**

The potentially important role of tributary feedbacks for buffering or accentuation of environmental signals (l. 63-66, l. 131-132) appears particularly important to me. I would suggest to emphasize it further, and especially to highlight the broader impact to the entire sedimentary system. Maybe you could build a case of how the effects of tributaries could strengthen or weaken the dynamics described by Simpson and Castelltort. That article is well known and I think that it would make your work even more approachable to the reader.

> *Thanks for pointing this out. We have added few lines in the introduction and discussion which point to the importance of these feedbacks and interactions for the whole sedimentary system, in connection with the work and results of Simpson and Castelltort (lines 64-65 and 860-865 in the manuscript version with changes).*

**Motivation for the flume setup.**

Somewhere in the text, maybe in a new section 2, the target landscape of the experiments should be spelled out. The flume seems to be representing the following fluvial landscape: two transport-limited streams (one twice as large as the other) with the same grain size join in a broad alluvial valley/floodplain of unlithified/uncemented sediments. The tributary builds an alluvial fan in the trunk channel. For the case of junctions between alluvial streams of the same order of magnitude Qw and same grainsize I would not expect the growth of an alluvial fan. The cases I have in mind where a tributary alluvial fan disturbs a main trunk are higher upstream. Paradigmatic would be the Illgraben Fan growing in Rhône Valley and constraining its river flow. In this case and the many others I can remember, there is an important grain size difference. I think I simply don't have the right references. I suspect that many readers may share the same experience as me. It would therefore be useful to discuss some field sites where the flume setup would apply. Preferably some that were studied for that dynamic.

> *We have added the description of the represented landscape in the method section (3.1).*
>
> *We understand the point raised by the reviewer and it is true that this setting may be peculiar of some specific region, as it may be the case of some catchments in the arid regions of north-eastern Argentina. There, thanks to several clast count measurements, we have evidence of jointly rivers draining alluvial material and carrying similar grain sizes (e.g. the Yacorite river joining the main Rio Grande in the Jujuy province of north-eastern Argentina). The tributary shows remnants of a paleo alluvial fan, suggesting that sometime in the past the Qs or Qw discharge of the tributary where different from those of today. However, the rivers have not been studied for the purposes analyzed in this*

*paper. Additionally, in most cases when an alluvial fan builds up in a main channel, the grain size distribution of this latter system is expected to change, as the channel slope adjusts to the incoming material brought by the tributary. It is clear that our examples represent a simplification of what may happen in natural settings, where the parameters that enters into play are many more than those used in the experiments. This is indeed a limitation inherent of our flume study. We have added a paragraph (5.4) on experiment limitations where we discuss, among others, also this aspect of the experiments and hope to accomplish to the point raised by the reviewer.*

**Representativity of each model run**

There misses a discussion of the relevance each individual run for the scenario explored. As detailed at length, alluvial systems have rich dynamics with a lot of stochastic processes. How confident are the authors that each run is a representative unique outcome of the scenario tested and not one of a wide range of possible evolutions? I fully understand that this is an inherent limitation of flume studies as each run represents tremendous work, but it would strengthen the framework if this limitation is directly addressed in a short paragraph.

*We agree with the reviewer and we discuss this limitation in the new paragraph 5.4.*

**Line by line**

• l. 121-130 The experimental work by Bonnet and Crave (Geology, 2003) on directionality of perturbations in landscapes would be particularly relevant for this paragraph.

*We thank the reviewer for pointing this out. We have added a sentence to include the reference to the work of Bonnet and Crave.*

• l. 254 It may be good to explicitly write that the level of the water sill is fixed.

*Done.*

• l. 269 I would suggest to point to Table 1 at the end of the first sentence already.

*Done.*

• l. 278-279 This seems a tall order to me. There is a lot of stochastic and non-linear processes in such a system. Wouldn't adding its parts yield more than their sum? Is there a reference for the feasibility of this?

*Yes, true. We cannot be sure that other processes do not interact. We have removed the sentence.*

• l. 333-335 This sounds more like the quantification of "straightness" rather than symmetry. The latter implies features within the floodplain to me. maybe add "axial" symmetry? this would make the link with the source-to-outlet straight line clearer.

*Done.*

• l. 367-369 For clarity's sake. V is then the volume of all sediments that were moved in the time interval, regardless whether they exited the section or not. It is the summed volume of all parcels of sediment mobilized during the interval, whether observed as new deposit or as new

erosion. However, any sediment bypass would not count toward V regardless of its sediment throughput. I think that this is what I understand from the text.

*Yes, this is correct.*

• l. 381 "deposited"? as in incised and deposited.

*Yes, changed.*

• l. 385, l. 389-390: How long is the spin-up phase? Is it 300 minutes after which the changes are observed (Figure 4)? And the spin-up phase is the complete adjustment to boundary conditions, correct?

*The spin-up phase represents the initial adjustments from the hand-made channel shape. Its timing changes from run to run and we have added a column to Table 1 where we stated, for each experiment, its total length and the spin-up time. After the spin-up phase the channels adjusted to the boundary conditions.*

• l. 546 "mainly" how can the valley widen in other ways than bank erosion?

*True, we have removed the word.*

• l. 557 "once the tributary reached equilibrium": from a slope perspective? It would be useful to restate whether it was after incision or aggradation.

*Yes, from a slope perspective. We have clarified it in the text. We are discussing here the T_NC1 experiment, so the system adjusts to the initial boundary conditions.*

• l. 569-570 Is this change in sediment mobilisation that visible in Qs_out? Or is the lack of tributary Qs merely replaced by main channel Qs during transient phase?

*Yes, the lack of Qs from the tributary is offset by the increased Qs in the mainstem from incision of the upper section. Therefore, the changes occurring in the tributary are not that visible in the Qs_out of the middle section. However, we do observe the delay in sediment transfer looking at the DoD figures (now moved to the supplementary material). There, we can observe that when the perturbation starts, sediment is initially deposited at the fan head and only with time is moved towards the main channel.*

• l. 577-578 "blocked" what is the exact meaning of blocked? Does it mean that 100% of the upstream sediment flux is effectively blocked, or that the sediment flux is limited and part of it is deposited?

*The second. We have added the word "partially" to clarify it.*

• l. 592-593 What kind of deposits are we talking about here? The material buried underneath the floodplain or terrace deposits where available?

*When possible, all of them. The more information available, the better incision and deposition histories can be reconstructed.*

• l. 684 one "r" is missing in prograde.

*Correct. Thanks.*

• l. 702-704 The dynamic of that competition must be heavily influenced by the respective erodibility of fan and bank. I imagine that a balanced situation like this one is rare. Tributaries often carry coarser sediment than the floodplain of the main channel. Or conversely floodplain material can be significantly consolidated and much harder to erode than loose fan material. Not even mentioning bedrock-lined valleys. It might be worth discussing comparisons with field examples again here.

> *We guess that with "balanced situation" the reviewer refers to all settings where two rivers flow on an alluvial plain. Although our set-up may resemble this type of landscape, we do not actually described a "balanced situation". We observed that a perturbation in the system produced a response those prevailing effects depended on the relative "strength" of the two rivers and the competition between them. In this context, when the tributary is prevailing the main channel gets deflected more, whereas when the main channel is "stronger", it manages to have a more straight path. Of course it is a simplification. There are many aspects that cannot be taken into account when working with lab- experiments, as it may be the case of different erodibility between fan and main channel or the presence of vegetation. Although they can change the dynamics of the system and the mechanisms with which sediment is moved, we could not evaluate their impact with our experimental setting. This has also been added in the limitation section.*

• l. 780 how? where?

> *Data will be made available through the Sediment Experimentalists Network Project Space to the SEAD Internal Repository and will possibly be accessible by the end of February 2020.*

**Figures**

• Figure 4: This is a very important figure but it is unfortunately hardly readable. Most profiles overlap and any pattern of change is almost impossible to decipher.
Have the authors tried to subtract the elevation along the average slope of the first profile from all profiles? This detrended curve would allow to spread the plots in the vertical. Further, the colour scheme is most likely not colour-blind friendly and should be amended (see Crameri's scientific colour scales for example).

> *We see the point. We have changed the figure following the reviewer's suggestion (each profile now plots with a scatter in elevation and is shown against the first-profile's average slope profile). However, we also kept the original plots to not lose the information about the changes in elevation. We also changed the color scheme, and Figure 5 and 6 (now Figure S1 in the supplementary material) accordingly.*

• Figure 7: the small outlines of the fan shapes is a great idea!

> *Thanks!*

• Figure 12: typos in "decoupling". The figure would be much stronger if examples from the field were listed to anchor these cases in a familiar context. What about aggrading main channel? Where does this setting fall?

> *Thanks for the typo. We understand the point of the reviewer but, considering that this figure is already very rich and contains a lot of information, we would prefer to not add*

*extra information on it. However we could add some examples if the reviewer strongly believes that it will be an added value for the manuscript. Nevertheless, we would like to point out that we are not aware of studies that have specifically analyzed the information reported in this paper, so that examples of field-cases would not really match the information reported here. Indeed, we explored the interactions between a tributary and a main channel and how this interplay may affect the transfer of sediment. This represented a knowledge-gap that may hinder important information for the reconstruction of climatic or tectonic histories of a certain region. Here, we provided a theoretical framework that may help filling this gap. It will be the readers who would need to see how our results may fit their own field site and up to which level they can use our framework for their analyses.*

*The case of aggrading main channels has not been tested in our experiments.*

Good luck to the authors for the revisions,

*Thanks!*

Best wishes, Luca Malatesta

---

## Author Comment (AC3) · 19 Feb 2020

[revised manuscript text omitted]

**(a) Fan aggradation**

*Non-influential alluvial fans*

Partial coupling
between FAN & MC

Complete coupling
within MC

**(b) Fan aggadation**
& progradation
*Influential alluvial fans*

Partial coupling
between FAN & MC

Partial decouping
between upper
and lower MC

**(c) Fan incision**
& progradation
*Influential alluvial fans*

Complete coupling
between FAN & MC

Partial decouping
between upper
and lower MC

on the FAN: top-down incision and healing wedge.
Erosion of material previously deposited.
Sediment raches the MC in pulses and
may record the perturbation
onset.

On the FAN: top-down deposition and propagation of $Q_{s\_in}$ signal.
Sediment reaching the MC records the perturbation
onset with a delay.

Fan elevation
new topography
old topography
Fan length

Upstream deposits:
increase sed. rate

Downstream deposits:
FAN supply +
Upstream dynamics

MC elevation
tributary
MC length

Upstream deposits:
Upper MC dynamics

Downstream deposits:
FAN supply +
pulses of upstream erosion

MC elevation
tributary
MC length

Upstream deposits:
increase sed. rate

Downstream deposits:
pulses of either FAN erosion
or upper MC erosion +
**bank contrib.**

MC elevation
tributary
MC length old topography
new topography
Fan elevation
Fan length

**(d) Main channel incision**
*Non-influential alluvial fans*

Complete coupling
between FAN & MC

Complete coupling
between upper
and lower MC

On the FAN: bottom-up incision and
erosion of material previously deposited.
Sediment raching the MC is immediately
transported away.

old topography
diffusive
knickzone
temporary
healing wedge
incision
in MC
new topography
Fan elevation
Fan length

Upstream deposits:
Upper MC erosion

Downstream deposits:
mixture of fan and
upper MC erosion +
bank contrib.

MC elevation
tributary
MC length

[revised manuscript text omitted]